# Worldwide Research Trends on Milk Containing Only A2 β-Casein: A Bibliometric Study

**DOI:** 10.3390/ani12151909

**Published:** 2022-07-27

**Authors:** Lucía Jiménez-Montenegro, Leopoldo Alfonso, José A. Mendizabal, Olaia Urrutia

**Affiliations:** IS-FOOD Institute, School of Agricultural Engineering and Biosciences, Public University of Navarre (UPNA), Campus de Arrosadia, 31006 Pamplona, Spain; lucia.jimenez@unavarra.es (L.J.-M.); leo.alfonso@unavarra.es (L.A.); jamendi@unavarra.es (J.A.M.)

**Keywords:** milk proteins, beta-casein, A2 milk, BCM-7, bovine

## Abstract

**Simple Summary:**

A1 β-casein has been correlated with adverse health outcomes, and, as a consequence, milk containing only A2 β-casein has emerged on the market. There has been a relevant increase in publications in this area since 2010. Food Science Technology and Agriculture were the main research areas of this topic. The term β-casomorphin was the most frequently used. The USA, New Zealand, and Australia were the most productive countries, though the most productive research institutions were, in absolute terms, from India, France, and Germany. The majority of the most cited studies that refer to A2 β-casein and health were reviews, and a few clinical trials have also been published.

**Abstract:**

The protein fraction of β-casein may play a key role in the manifestation of a new intolerance: milk protein intolerance. The most common forms of β-casein among dairy cattle breeds are A1 and A2 β-casein. During gastrointestinal digestion of A1 β-casein, an opioid called peptide β-casomorphin-7 (BCM-7) is more frequently released, which can lead to adverse health outcomes. For that reason, novel products labelled as “A2 milk” or “A1-free dairy products” have appeared on the market. In this context, a bibliometric analysis on A2 β-casein research was carried out through the Web of Science (WoS) database. The main objective of this work was to provide an overview of the state of the art in the field of β-casein A2 by analyzing the number of publications per year, trends in thematic content, the most frequently used terms, and the most important institutions and countries in the field. This bibliometric study showed that a greater effort is needed to determine the possible implications of this novel product for human health and the market.

## 1. Introduction

Milk is a very important food resource for a large number of people [1,2]. It is obtained from the mammary gland secretions of mammals, and is mainly composed of water (around 86%), but also contains lactose sugars, triglycerides, high-quality proteins, minerals (calcium, magnesium, selenium), and vitamins (riboflavin, vitamin B12, pantothenic acid) [2,3,4]. In bovines, milk proteins fall into three categories depending on their solubility potential: caseins (at around 80%), whey proteins (about 14%), and fat globule membrane proteins [4].

Milk from domestic dairy mammals has been part of the human diet for thousands of years. However, in recent years, with the increase in production and consumption of dairy products, human health problems related to cow’s milk protein allergies and milk intolerances have increased [2,5]. Cow’s milk protein allergy (CMPA) is an adverse immune reaction that occurs when the body comes into contact with dietary antigens in cow’s milk [5]. Because of this, donkey and horse milk have been used as an alternative food for children suffering from CMPA [6]. Among milk intolerances, the most common is lactose intolerance, which is caused by a reduced or absent activity of the lactase enzyme, leading to lactose malabsorption [7]. To solve this problem, “lactose-free milk”, which is a lactose-free commercial dairy product, has been marketed.

In the past few years, researchers have studied the possible implications of the composition of the protein fraction of β-casein for the manifestation of a new intolerance: milk protein intolerance. Many people who consider themselves lactose intolerant may not actually have this problem [5]. Caseins account for approximately 78% of the total protein in bovine milk and, within them, β-casein represents 30% [8,9,10]. Dairy cattle have 12 β-casein variants (A1, A2, A3, B, C, D, E, F, H1, H2, I, and G) [11]; however, only seven of these (A1, A2, A3, B, C, I, and E) have been detected in European cattle breeds. The A2 variant is considered the oldest variant, from which the others originated via mutation. The most common variants are A1 and A2; the B variant is less common in most breeds [12,13]. The β-casein produced by dairy cows in their milk is controlled by a single gene, located on chromosome 6, called *CSN2* [11]. The A1 β-casein differs from the A2 β-casein by a different amino acid at position 67 of the peptide chain; this is histidine (His67) in the A1 variant and proline (Pro67) in the A2 variant. His67 within A1 has been associated with an increased susceptibility of A1 β-casein to enzymatic hydrolysis by gastric enzymes such as pancreatic elastase, pepsin, and leucine aminopeptidase [14,15]. These changes in digestion patterns associated with A1 β-casein lead to the release of BCM-7 during gastrointestinal digestion. BCM-7 is an opioid peptide that may have biological actions, including binding to opioid receptors in neuronal and non-neuronal tissues [16,17]. In this sense, BCM-7 could be responsible for possible adverse health outcomes associated with A1 β-casein consumption, giving rise to milk protein intolerances [18,19,20]. In contrast, A2 β-casein consumption could reduce gut-related discomfort [15]. In this sense, a growing range of dairy products labelled as “A2”, “A2 milk”, or “A2 beta-casein protein” have emerged on the market. These novel products, which contain only A2 β-casein protein, may be of interest to the milk protein intolerance population. Moreover, A2 cow’s milk may provide a diversification option for family-sized and small farms in the milk market, although possible consequences for milk production and technological characteristics should be considered [21,22,23].

In this context, a deep bibliometric study using the Web of Science database was developed. Bibliometrics is a statistical tool for obtaining consistent data from the literature that describes the institutions, researchers, and other sources of interest. Moreover, the use of bibliometric indicators to measure the contributions of researchers, countries, and organizations in the field of knowledge can be considered an economic and social indicator since they are investing a lot of resources [24,25,26]. Thus, the objective of the present study was to determine the state of research in the field of milk or dairy products containing only the A2 β-casein variant. The information provided by this methodology enabled us to comprehend, from a viewpoint supported by the literature, scientific trends in this specific field, guiding future directions of research.

## 2. Materials and Methods

There are a wide range of bibliometric databases used for bibliometric analysis, such as Web of Science (WoS), Microsoft Academic, Google Scholar, and Scopus. First, we undertook a general overview with the search query of interest among some other major databases, such as Scopus and PubMed, but the number of papers on β-casein A2 retrieved was significantly lower than we obtained through WoS. For example, in the Scopus database with the search query “beta-casein A1 OR beta-casein A2”, we only obtained 260 papers, so we decided to choose the WoS database, as a large number of scientific publications were available on this issue. Moreover, it allowed us to download data more easily for bibliometric purposes and offered robust tools for analyzing scientific publications [27].

The search was conducted in February 2022 to compile scientific publications containing the search query in the title, abstract, and/or keywords of the documents. The search was conducted in “all databases” available in WoS, and not only in the “WoS Core Collection”. A first general search was conducted using the term “beta-casein”. On this point, it should be mentioned that using the term “β-casein” in this first search would have given a low number of publications because the database did not detect the character “β” as “beta”. Subsequently, a second more specific search was carried out using the search query “A2 beta-casein OR A1 beta-casein” in order to obtain scientific publications related to milk containing only the A2 β-casein variant or A1 β-casein-free milk. At an early stage, a specific second search with the term “A2 milk” was considered, but this was dismissed because the database did not develop the search correctly, as scientific publications with exclusively the term “milk” or the term “A2” were also included in the searched documents. Both searches were restricted from the first publication year to 2022.

The publications obtained from the search query were categorized according to the following aspects: number of publications per year, research area, countries, affiliations, document type, and sources (journals). Datawrapper is a product that allows the visualization of information through charts, maps, and tables, and in this study was used to make a world map using different colors depending on the number of publications on this topic available in each country. Furthermore, VOSviewer, which is a computer program used for displaying large bibliometric maps in an easy-to-interpret way [28], was used to analyze the keywords associated with the entire bibliographic analysis in order to achieve a rapid visualization of the core content.

The number of times that a scientific publication is cited reflects the relative impact that a single publication has on the entire scientific community. Two different indexes, both used in previous bibliometric studies [25,29], were used to evaluate the impact of each source on the A2 β-casein research field. The h-index is a standard metric defined as the maximum value of h, such that the given author or journal has published at least h papers that have been cited at least h times [30]. This index allows us to study the impact of a journal, an institution, a scientific paper, or a country [31]. Another metric that has been commonly used in other bibliometric studies and is available on WoS is the 5-Year Impact Factor generated by Journal Citation Reports (JCR) [32]. The JCR is used to measure the professional quality of journals and is based on the number of citations that each publication has received in the journal in question [30]. The most 50 cited articles focusing on A2 β-casein and health problems were analyzed to try to find out if a correlation between A2 β-casein positive results and the number of citations existed. A flow chart of the search approach is shown in Appendix A.

## 3. Results and Discussion

### 3.1. Evolution of Scientific Output

A total of 10,408 documents with “beta-casein” in the titles, abstracts, or keywords were retrieved from the WoS database. It is remarkable that no β-casein-related documents appeared until the year 1942, and the number of documents was very low until the 1960s (Figure 1). Since then, a general increase in the number of publications has occurred, but with several fluctuations throughout the years. In the 1990s, there was a sudden increase in the number of β-casein related documents. After that decade, the trend was of a progressive increase, with some decreases in some years that could be linked to the changes taking place in the agricultural and food sectors [33].

The search query including the terms “A2 beta-casein OR A1 beta-casein” in the title, abstract, or keywords retrieved 618 documents, a considerably lower number of documents (5.93% of the total β-casein related documents). This low percentage was likely due to the fact that it was not until the late 1990s that the number of written documents on this topic exceeded 10. This indicates that A2 β-casein and A2 milk research is a very novel issue. It was not until the 2010s that an increase occurred, and reviews of the implications of β-casein variants for human health were published [3,18,19,34]. During the years 2020 and 2021, the maximum number of publications on this topic was reached [14,35,36,37].

### 3.2. Distribution of Scientific Output in Subject Categories

Based on the WoS classification, the distribution of publications in the β-casein research field covered a total of 121 subject areas. However, only 20 areas included more than 500 publications on β-casein (Figure 2).

The largest number of documents were from *Food Science Technology* (7343 documents), *Biochemistry* (6287 documents), and *Agriculture* (6133 documents). It was remarkable that, from a total of 10,408 documents, many were included in more than one research area. In this sense, for example, if the search was restricted to documents only included in *Food Science Technology* and *Agriculture*, a total of 7715 documents would have been retrieved, meaning that a lot of documents in the β-casein research field were included in both fields by the WoS database. Within these two research areas, different aspects of the β-casein research field were considered. Some of them made reference to the antigenic structure of β-casein, and others mentioned either the ability of β-casein protein to serve as a cloning vector or the capacity of β-casein gene promoter to allow heterologous gene expression in genetic engineering [38,39]. In contrast, there were also other studies that referred to the effects of β-casein variants on milk intolerance or analysis of β-casein alleles in mammals such as bovines, camels, donkeys, or caprine [15,40]. Apart from these three main research areas, there were other important research areas, such as *Chemistry* (2912 records), *Genetics Heredity* (2897 records), *Nutrition Dietetics* (2665 records), and *Physiology* (2657 records). The rest of the research areas shown in Figure 2 included less than 2000 records each.

Distribution by area was similar when the publications only referred to the A2 β-casein research field, and the most frequent research areas were *Food Science Technology* (525 records), *Agriculture* (443 records), and *Biochemistry* (333 records). However, in this case, the number of publications retrieved in *Agriculture* was higher than in *Biochemistry*, contrary to the results observed in the β-casein research field. This indicated that the focus of research in the area of A2 β-casein was in *Agriculture, Livestock,* and *Food*. Numerous papers focused on determining β-casein allele frequencies in different cattle breeds by identifying the β-casein gene (*CSN2*) or β-casein protein [41,42,43,44]. In addition, many papers discussed the influence that milk casein polymorphisms may have on milk yield, milk protein content, or milk fat yield [45,46]. Other papers revolved around determining the potential nutritional and health implications of the A1 and A2 β-casein variants in human or animal populations [10]. For all these reasons, other important research fields in A2 β-casein were *Genetics Heredity* (281 records) and *Nutrition Dietetics* (213 records). It should be mentioned that a lot of documents were included in more than one research area by the WoS database.

In Figure 3, the number of A2 publications in each research field was relativized with respect to the total number of publications in each research field. The contribution and importance of A2 β-casein-related publications has undergone changes. It was possible that the number of publications in A2 β-casein in a specific research field was very high, not due to the high importance of this research area in A2 β-casein research, but because, in general, a large number of documents on other topics were published in that research area. This was the case in the *Biochemistry Molecular Biology* research area, which was the third most frequent source of A2 β-casein-related publications, with 333 records (Figure 2), but the relative importance of the A2 β-casein research field in *Biochemistry* was lower and accounted for 4.36% of the total records in *Biochemistry* (Figure 3). The top 10 most important research areas for A2 β-casein research were *Food Science Technology* (26.15%) was placed, followed by *Agriculture* (13.97%), *Nutrition Dietetics* (12.94%), *Genetics Heredity* (6.78%), *Reproductive Biology* (5.04%), *Biotechnology Applied Microbiology* (4.98%), *Biochemistry Molecular Biology* (4.36%)*, Veterinary Sciences* (4.09%), *Physiology* (3.67%), and *Endocrinology Metabolism* (2.76%).

### 3.3. Analysis of Terms Used in Titles and Abstracts

To identify trends in the A2 β-casein research field, the VOSviewer computer program was used to determine the most important terms (keywords) in this research topic. Although this program is suggested for larger analyses, with the recommended number of documents being between 1000 and 5000 [28], it can also be used to perform smaller analyses. A general analysis with many terms usually includes terms that appear more than 100 times [32], but in this analysis, in which fewer terms were analyzed (618 records), a minimum occurrence of 12 times was used. In addition, when working with terms, the definition of *Occurrence* (the number of documents in which a keyword occurs) changed; it depended on the counting method selected in the VOSviewer program. In the case of binary counting, the *Occurrence* attribute indicates the number of documents in which a term appears at least once. However, in the case of full counting, the meaning of *Occurrence* refers to the total number of occurrences of a term in all documents retrieved [28]. For the present bibliometric study, which included a low number of publications, there were hardly any differences between a binary counting map and a full counting map, therefore, a *Full Counting Term Map* only was carried out (Figure 4). From the total terms included in the title and abstract fields (9942 terms), terms that appeared 12 or more times were selected (286 terms). Moreover, a relevance score and the 60% most relevant terms were calculated using the VOSviewer program, giving 172 as the final number of terms selected for the *Full Counting Term Map.* The names of some terms described below are not presented in Figure 4, since only the most relevant or significant ones for the VOSviewer program were included. In the term map, the circle size under the word is directly proportional to its frequency of occurrence in the literature [28].

In the first cluster, shown in red, the terms “peptide”, “beta-casomorphin”, “activity” and “gastrointestinal digestion” were very frequent terms. This cluster was mainly related to the opioid peptide β-casomorphin (BCM). The terms “casein fraction”, “cleavage” or “release” were also present because BCMs are released from the 60–70 sequence of the β-casein fraction [14]. BCMs become biologically active during gastrointestinal digestion or milk processing [14,47] and are obtained from various sources of milk, such as bovine, buffalo, and sheep, but not from goats and camels [48]. Human milk also contains BCMs, which are expressed in the colostrum and in mature milk [14]. For that reason, the terms “colostrum”, “mature milk”, and “human” also appeared in the term map. The presence of BCMs in “cheese” (another important term in the map) depends on the milk treatment process, pH, coagulant, and strain starter used, and the temperature and humidity employed during ripening [14]. Among BCMs, in the A2 β-casein research, BCM-7 is especially important, being more frequently released by A1 β-casein [49]. BCM-7 can influence the nervous, endocrine, and immune systems by activating the gastrointestinal tract’s µ-opioid receptors [20]. Thus, BCM-7 is thought to be responsible for potential adverse health-related outcomes associated with A1 β-casein consumption, and it has been widely discussed in the literature [8]. In the term map, the term “alpha” was also very common and could refer to other casein subtypes: alpha (α)-S1-casein (15–18%) and alpha (α)-S2-casein (11%), to the whey protein alpha (α)-lactalbumin [12,14], or to other terms which included the term “alpha” in the name. Furthermore, the term “gamma-casein”, which is the degradation product of casein, was also very frequent [14]. The term “retinoic acid” was also recurrent, since retinoic acid has been shown to induce differentiation of MAC-T cells (mammary alveolar cells), and this causes an increase in mRNA expression of the α-S1-casein, α-S2-casein, and β-casein genes [50].

In the second cluster, shown in green, terms related to milk were very common, such as “milk consumption”, “milk product”, “cow milk”, and the relationship between milk consumption and health-related outcomes could be observed through the presence of terms such as “diabetes”, “schizophrenia”, “cardiovascular disease”, “disease”, “heart disease”, and “inflammation”. The term “beta-casomorphin” (BCM) was located in the red cluster, but was also next to the green cluster, with a lot of linkages with their terms. This was probably because a high number of reviews on the A1 and A2 β-casein research field have focused on the description of the relationship between BCM-7 and possible health-related adverse effects, such as gastrointestinal problems [9,49], heart disease, diabetes, or autism [10,20,51,52]. However, the number of clinical trials in humans remains low [18,19], thus they cannot be considered conclusive and further studies are required to provide further insight into the possible effects of BCM-7 on human health [14,36]. The term “lactose intolerance” was also very frequent because this milk intolerance is the most common in the human population [5]. In recent years, in contrast, studies of the possible implications that the protein fraction of β-casein may have for milk protein intolerances have been recurrent.

In the third cluster, shown in blue, the terms “frequency”, “genotype”, “population”, and “csn2” were the most frequent because many studies in the field of A2 β-casein research have focused on genotyping cattle breeds to determine the genotypic frequency of the *CSN2* gene. Among these terms, “dairy cattle”, “allele”, “a1a1”, “a1a2”,” a2a2”, and “a1 allele” were also recurrent terms. The variety shown among genotypic frequencies for the *CSN2* gene in cattle breeds worldwide may be a reflection of local breeding policies and some cross-breeding, most likely aimed at increasing milk production traits [41]. The terms “Simmental”, “Jersey”, and “Holstein” were common in the search query, because assessment of A1 and A2 variants of the *CNS2* gene has been regularly developed in these breeds [46,53]. The terms “India” and “Sahiwal” also appeared in the term map, because in India, the Sahiwal breed is considered one of the best dairy cattle breeds, and it has been widely used to develop crossbred cows and to determine β-casein polymorphisms [54,55].

In the fourth cluster, shown in yellow, the terms “kappa-casein” (K-casein) and “beta-lactoglobulin” (β-lactoglobulin) were the most frequent, since both milk protein polymorphisms have been frequently studied together with A1/A2 β-casein polymorphisms. The analysis of the relationship between these milk protein genes, which are known as major genes, and milk production traits (milk yield, fat content, fat yield, protein content, and protein yield) has been frequently developed [45,46,53,56,57,58]. These major genes are *CSN1S1* (α-S1-casein), *CSN1S2 (*α-S2-casein), *CSN2* (β-casein), *CSN3* (K-casein), and *BLG* (β-lactoglobulin). Some studies have detected significant associations between the *CSN1S1-CSN2-CSN1S2* haplotype block with both milk protein and milk yield [56], whilst a recently developed meta-analysis suggested no relationship between any milk production trait and the *CSN1S1* and *CSN2* genotypes [46]. This latter study also detected that some *CSN3* genotypes and *BLG* genotypes are associated with higher fat and protein content, suggesting that these major genes will be useful markers for improvement of those traits. In this manner, terms such as “csn3”, “milk yield”, “fat yield”, “protein content”, “protein composition”, and “haplotype” were recurrent in the term map. The term “b allele”, which appeared numerous times, could refer to different milk protein polymorphisms. For example, in one study it was established that the BC haplotypes of (α)-S1-casein, B and A1 β-caseins, B K-casein, and B α-lactoglobulin were favorable for rennet coagulation [45]. The term “snp” and “exon vii” were present in this cloud since A1 and A2 β-casein differ by a Single Nucleotide Polymorphism (SNP) in the *CSN2* gene [11,14]. These SNPs are SNPs of *CSN2* exon VII allele A2 (201-CCT-203, GenBank: JX273430.1) and SNPs of *CSN2* exon VII allele A1 (201-CAT-203, GenBank: JX273430.1).

### 3.4. Publication Distribution by Country and Institution

A2 β-casein-related studies have been developed in 57 different countries, since it constitutes a very novel and specific research field. Among the top 20 most productive countries in this research were: the United States of America (U.S.A.) (48 records), New Zealand (47 records), Australia (40 records), Germany (39 records), Italy (35 records), India (32 records), France (25 records), China (21 records), Denmark (15 records), Spain (15 records), the United Kingdom (15 records), Finland (13 records), Poland (12 records), Canada (11 records), Mexico (10 records), Russia (10 records), Brazil (9 records), South Korea (9 records), Sweden (9 records), and the Netherlands (8 records) (Figure 5).

In relation to Figure 5, the U.S.A. was the country with the most publications, and this could be due to several reasons. Firstly, because of its size, population, and the number of affiliations and sources, the U.S.A tends to be at the top of any research area worldwide. The predominant cattle breed in the U.S.A. is the Holstein (92%), and this breed produces milk containing around 66% A1 β-casein. In contrast, other breeds such as the Guernsey, which produces milk with high quantities of A2 β-casein (90%), represent a small percentage of U.S.A. dairy cows [51]. Both types of milk are pooled and, as a consequence, the majority of the U.S.A. population consumes milk containing high levels of A1 β-casein, thus possibly increasing interest in determining the possible implications of this β-casein fraction for human health. Another reason that could explain the high number of publications in the U.S.A. may be that, in 2003, the New Zealand *A2 Corporation Ltd.* (which had worldwide rights for producing and marketing milk with only A2 β-casein) exclusively licensed patents and trademarks to the US-based Ideasphere Incorporation (ISI) to market A1-free products in North America [51,59]. Therefore, as a direct result of this, an increase in the interest in A1 and A2 β-casein research studies and the evaluation of the potential impact of this novel product in the market would have been expected.

A large number of studies were performed in Australia and New Zealand, derived from the *A2 Corporation Ltd.*, established in 2003, with one principal site in Auckland (New Zealand) and another in Australia. Since that year, A2 milk has been sold in New Zealand and Australia as a premium brand with a natural protein content. In 2005, the company was renamed “The A2 Milk Company”. In this context, many studies have been developed, all with *A2 Corporation Ltd.* participation, suggesting that consumption of A2 milk may attenuate the gastrointestinal symptoms of milk intolerance [18,34,60]. Therefore, further studies with other perspectives are needed to determine A1 and A2 β-casein implications for human health.

In India, the first cow-milk-consuming country, there is a significant interest in characterizing β-casein polymorphisms among indigenous cattle breeds, and for that reason, a high number of publications have been carried out in this country (Figure 4). Genotyping results with different DNA-based techniques such as High-Resolution Melt (HRM) analysis [41] or allele-specific PCR [42] revealed that Indian native cattle breeds have a high frequency of the favorable A2 allele and so constitute a great resource to meet the global demand for A2 milk [61].

Furthermore, in Europe, a lot of studies in the A2 β-casein research field have been carried out. The main producers of cow’s milk are Germany (19%), France (15%), the United Kingdom (9%), the Netherlands (8%), Poland (8%), Italy (8%), Spain (5%), and Ireland (5%), which together account for three-quarters of the total EU production [62]. Significant interest therefore exists in this region, too, for adaptation to changes in the milk market and the promotion of innovative products, such as A2 milk. To understand the demand for this differentiated product, consumer choices and price competition need further analysis. Moreover, the identification of breeds and selection of animals for A2 milk production is important for public health and animal production. Among EU countries, especially Italy, there exists further knowledge of the occurrence and frequencies of β-casein variants and genotypes in the dairy cattle of this region [63].

Regarding affiliations, the most productive ones in the A2 β-casein research field are shown in Table 1.

The majority of affiliations were research centers in the fields of food and agriculture, as well as multidisciplinary sites, with universities playing a key role. At the top was the *Indian Council of Agricultural Research* (*ICAR*), which is an autonomous body responsible for coordinating agricultural education and research in India. ICAR was followed by *l’Institut National de Recherche pour l’Agriculture, l’Alimentation et l’Environnement (INRAE)*, which is a public research institute working for the sustainable development of agriculture, food, and the environment in France. It is noteworthy that these first two affiliations are both related to agricultural and food science, in accordance with the fact that the two first research areas among A2 β-casein publications were *Food Science Technology* (525 records) and *Agriculture* (443 records) (Figure 2). The rest of the affiliations shown in Table 1 are universities. Among them, the *Justus Liebig University Giessen* in Germany is in third place, followed by the *University of Melbourne* (Australia), the *University of Milan* (Italy), the *University of Warmia* (Poland), *Aarhus University* (Denmark), *Université Paris-Saclay* (France), *Lincoln University* (New Zealand), and finally the *University of Brescia* (Italy). The *University of Warmia* can be highlighted for its role as a center for the Russian language and culture, and for being one of the top medical universities in Poland. These results are in accordance with the contribution of each country to A2 β-casein research. For instance, despite the small population of New Zealand with respect to other countries, one affiliation of this country is among the top 10 most productive ones in A2 β-casein research, which is in accordance with the importance that New Zealand has in this field (Figure 6), and also with the fact that the *A2 Corporation Ltd.* was first established in this country.

In line with this, the relative importance of affiliations in A2 β-casein research, taking into account the number of A2 publications with respect to the total number of publications of each affiliation, is represented in Figure 6. According to this analysis, the affiliation with the highest relative importance in the A2 β-casein research field was *ICAR* in India (40.51%), followed by the *Russian Academy of Sciences* in Russia (17.32%), *New Zealand Research Institute* in New Zealand (16.52%), *University of Warmia* in Poland (4.34%), *INRAE* in France (4.16%), *Justus Liebig University of Giessen* in Germany (2.84%), *Norwegian University of Life Sciences* in Norway (2.73%), *University of Brescia* in Italy (2.28%), *Lincoln University* in New Zealand (2.26%), and finally, *Université Paris-Saclay* in France (1.82%). In this analysis, some changes were observed with respect to Table 1. *The Russian Academy of Sciences* (7 records) and *the New Zealand Research Institute* (8 records) stood out in second and third position, respectively, in relative importance in the field of A2 β-casein research. This denoted that both affiliations were more specialized in this topic, and also that the results in Figure 6 may be a reflection of the real contribution each country has made to A2 β-casein research. In line with this, India, Russia, and New Zealand have had a large impact on A2 β-casein research. On the other hand, no affiliations were represented for China and the U.S.A., although in Figure 4 both countries appeared among the most productive, maybe due to the greater number of institutions and the relative lower importance of A2 β-casein studies. Apart from these affiliations, it is worth noting that *A2 Corporation Ltd.,* which is neither a university nor a research center, but a private company, has developed 11 studies, all of them related to A2 β-casein research.

### 3.5. Publication Distribution by Document Type and Source

Documents on A2 β-casein research recovered from the WoS database could be divided into nine document types (Table 2).

From the results obtained directly from the WoS database, some documents were in more than one category, thus we undertook a manual analysis so each document would be in only one. The most common document type was “article”, which accounted for 502 records and represented 81.2% of the total publications on this topic. Another important category was “review”, which accounted for 31 records (5.0%), followed by “patent” with 27 records (4.4%), “meeting” with 21 records (3.4%), “book” with 10 records (1.6%), and finally, “letter” with 8 records (1.29%). Additionally, 19 documents could not be placed in a category and were classified as “other” (3.1%). These results indicated that the majority of authors chose the article format as the most suitable option to publish their results and findings in the A2 β-casein research field. Nevertheless, a lot of authors selected reviews as a way to disseminate their scientific research in this field. Reviews constitute an important format to summarize important findings and studies of a specific research project.

Regarding the number of publications by source, nearly 618 records were found in 368 different journals, although the vast majority had published less than 10 records. A list of the top 10 journals is presented in Table 3.

It is important to highlight that other sources of information, mainly books and patents, were also important for this topic. Among the patents, the *PCT International Patent Application* and the *United States Patent Application* were important, with nine and six publications in A2 β-casein research field, respectively. The PCT is an international treaty with more than 150 Contracting States, and the United States Patent Application is one of the three major national and regional offices existing worldwide. The subjects of patents published in both journals in the A2 β-casein research field were mainly separation methods to obtain A1-free products, or protocols explaining the relationship between A2 β-casein and glucose levels and antioxidant capacity.

As shown in Table 3, after analyzing the number of publications, the *Journal of Dairy Science* (JDS) was the most productive journal in this research field (61 records). This may be because JDS is the leading dairy research journal in the world and publishes research related to the production and processing of milk or milk products intended for human consumption. This magazine was followed by the *International Dairy Journal* (18 records), the *Journal of Dairy Research* (16 records), *Animal Genetics* (13 records), and *Food Chemistry* (12 records), thus making it clear that in the research topic, the majority of journals referred to dairy products, animals, food technology, or nutrition. The first three journals included the word “dairy” in their titles, and the words “nutrients”, “nutrition”, “animals”, or “food” could also be found in the titles of the rest. It is worth pointing out that in this top 10 list of journals, none has a multidisciplinary character, possibly as a result of the particularity of the topic. The *European Journal of Biochemistry*, which was in 9th position in terms of number of publications (Table 3), was published from 1967–1998, when it was renamed the *Federation of European Biochemical Societies* (FEBS) in 2005.

Regarding the impacts of the scientific journals, other conclusions could be reached. Based on the 5-Year Impact Factor generated by Journal Citation Reports (JCR), which was available on WoS and reflects the professional quality of a journal during the last 5 years, *Food Hydrocolloids* with 9.147, *Food Chemistry* with 7.516, and *Nutrients* with 5.719 were the most important journals. Regarding the h-index, which is also used to measure the impact of a journal, *Food Chemistry* (262), *Journal of Dairy Science* (191), *Food Hydrocolloids* (159), and *International Dairy Journal* (140) were the most important journals for this topic. The h-index did not show results in accordance with those of the JCR, since the latter only takes into account the impact of the journal in the last 5 years and not the overall impact of the journal. Furthermore, the high impact factor of *Food Chemistry* and *Food Hydrocolloids* in both the JCR and h-index suggested that the impact of the papers in the A2 β-casein research field published in the food field was higher than in other fields.

### 3.6. The Most Cited Articles Involving A2 and A1 β-Caseins and Health-Related Implications

We also analyzed whether the majority of the results of the most cited studies in the A2 β-casein research field were in favor of A2 β-casein consumption or against it. The same search was performed: “A2 beta-casein OR A1 beta-casein”, and the option “sort by: citations: higher first” was selected, to select the most important studies with the highest impact on this research field. Among the results obtained, a high number of studies addressed the analysis of milk protein polymorphisms and their influence on milk technological characteristics (Appendix A), but this was not the main purpose. To remove such articles, only the articles and reviews on the possible health outcomes for animal and human populations associated with A2 or A1 β-casein consumption were selected manually. In Table 4, a list of the articles on health topics in the A2 β-casein research field with more than 50 citations is shown.

It is remarkable that 11 of the 12 most cited studies concluded that A2 β-casein consumption was desirable when compared to A1 β-casein consumption. The majority advocated A2 β-casein consumption over A1 β-casein, which they correlated with possible health problems among populations such as people with diabetes, cardiovascular diseases, gastrointestinal problems, or even schizophrenia and autism [64,65,66]. A few even argued in favor of A2 β-casein, saying that dairy products containing this variant were more easily digestible in the milk intolerance population [51,52,53,54,55,56,57,58,59,60]. These high-impact studies were mainly developed in New Zealand and Australia, in which is situated *A2 Corporation Ltd.* the main company worldwide in charge of commercialization of A2 milk and A2 dairy products, and a participant in 7 of the 12 studies in Table 4. The most cited study was titled “*Polymorphism of bovine beta-casein and its potential effect on human health”* (*n* = 177 citations); it was published in 2007 [12] at the University of Warmia (Poland). This affiliation was previously mentioned because in the A2 β-casein research field this university has high relative importance (Figure 6). The second most important study was titled “*Type I (insulin-dependent) diabetes mellitus and cow milk: casein variant consumption*” (*n* = 136 citations), and was developed in 1999 [64]. Although *A2 Corporation Ltd.* participation did not occur, it was developed by the University of Auckland, which is one of the company´s locations. The third most important study was titled “*Beta-casein A(1), ischaemic heart disease mortality, and other illnesses”* (*n* = 116) and was developed in 2001 [65], with the unique participation of *A2 Corporation Ltd.* As one conclusion of this analysis, a correlation between A2 β-casein positive results and number of citations could exist, because, with the exception of [68], none of the papers with more than fifty citations found a negative result for A2 β-casein consumption. However, reports from official institutions indicate that there is still not enough evidence to change dietary recommendations [71], nor to establish a cause-effect relationship between oral intake of BCM-7 or related peptides and the etiology of the diseases that have been suggested [72]. It is also remarkable that among the twelve articles with more than fifty citations, only three corresponded to clinical human trials. The most frequent type of publication was reviews of previously published papers; all of these indicate that no conclusive results have been found, or that more investigation is needed. The rest of the publications were field experiments (analyzing field data from available databases) and animal (in rabbits and rats) or laboratory experiments (simulating gastro-intestinal digestion).

## 4. Conclusions

After developing this bibliometric study, it was found that a large number of studies were developed either in Australia or in New Zealand, mainly due to the presence of the *A2 Corporation Ltd*. This was the author or sponsor of many studies on A2 β-casein and health implications, in which positive results for A2 β-casein consumption were found. Many of these studies were based on the widespread belief over the last few years that A2-related products are better than A1. Reviews were often carried out, and always came to the conclusion that A2-related products may be better than A1, but more investigation is needed. Although the number of publications has increased worldwide during the last few years, involving different research areas and institutions, the number of clinical trials in humans remains limited.

## Figures and Tables

**Figure 1 animals-12-01909-f001:**
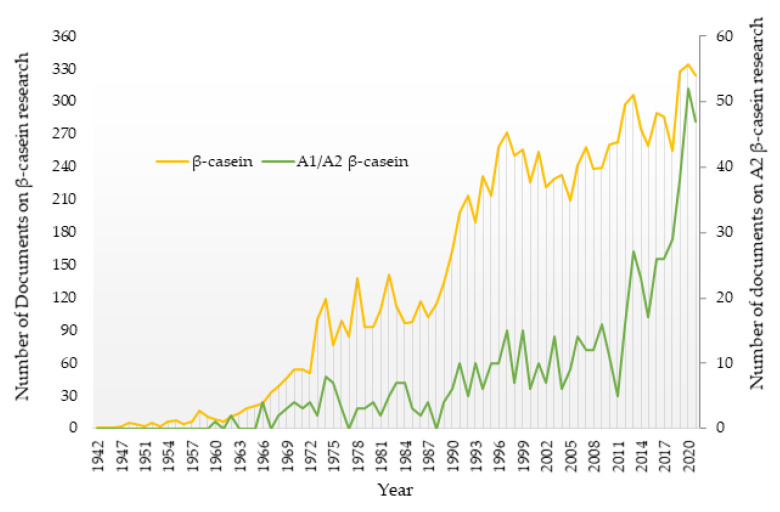
Trends in publications on β-casein research in absolute terms (yellow line) and in the A2/A1 β-casein research field (green line) in the period 1942–2021. Documents from the year 2022 were not included in the figure since the year was not ended at the time of writing.

**Figure 2 animals-12-01909-f002:**
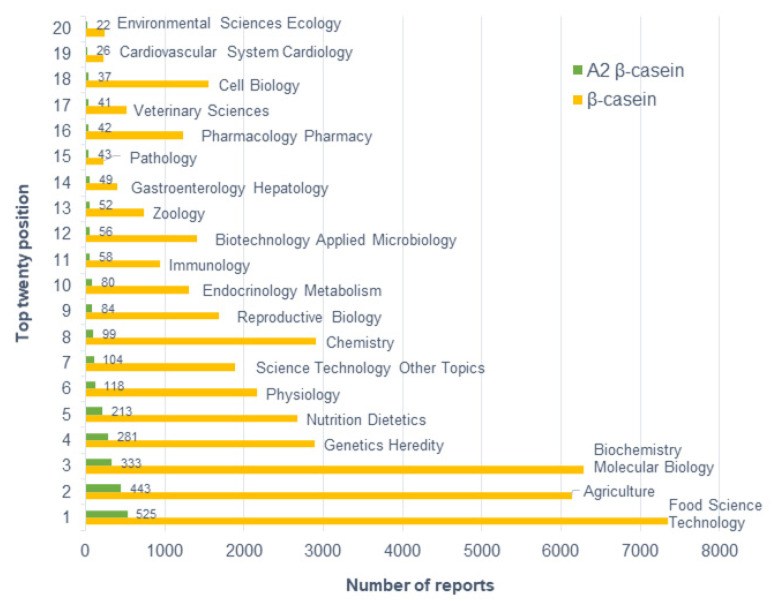
Distribution (number of reports) of worldwide research on β-casein and A2 β-casein by subject area, as classified by the WoS database.

**Figure 3 animals-12-01909-f003:**
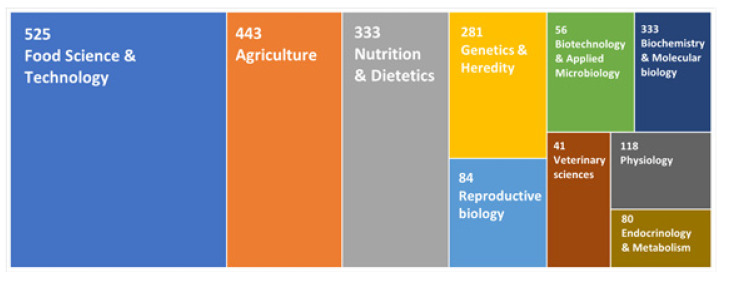
TreeMap chart of the most important research areas for the A2 β-casein field. The absolute number of publications referring to A2 β-casein in each research area is indicated. The areas of the chart are proportional to the number of publications with respect the total number of publications of each research area. Only the 10 most important areas in relative terms are shown.

**Figure 4 animals-12-01909-f004:**
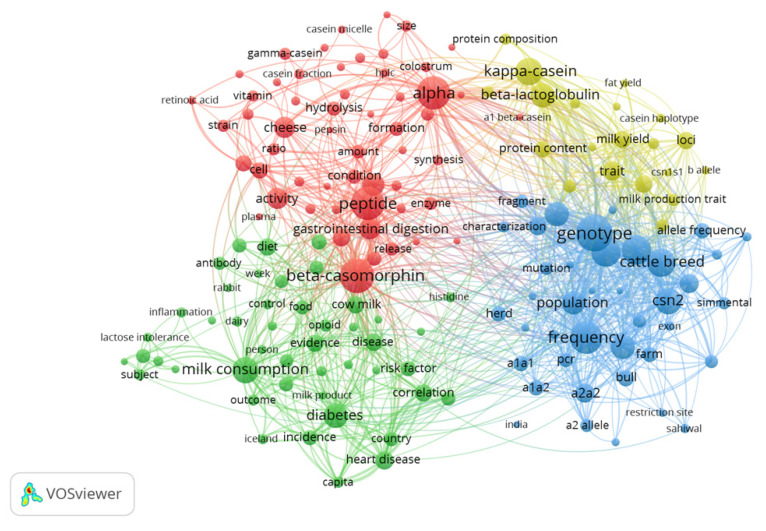
Terms clouds based on worldwide A2 β-casein research. Terms used more than 12 times in titles and abstracts. The size of the circle under the word indicates the term frequency. The four clusters represent the relationship between the items. Each cluster is represented with a different color by the VOSviewer program.

**Figure 5 animals-12-01909-f005:**
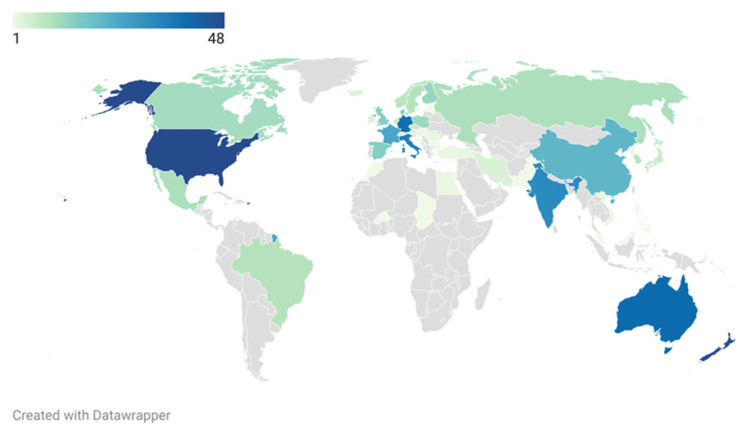
Contribution by country to research documents on the A2 β-casein research field in the period 1960–2022.

**Figure 6 animals-12-01909-f006:**
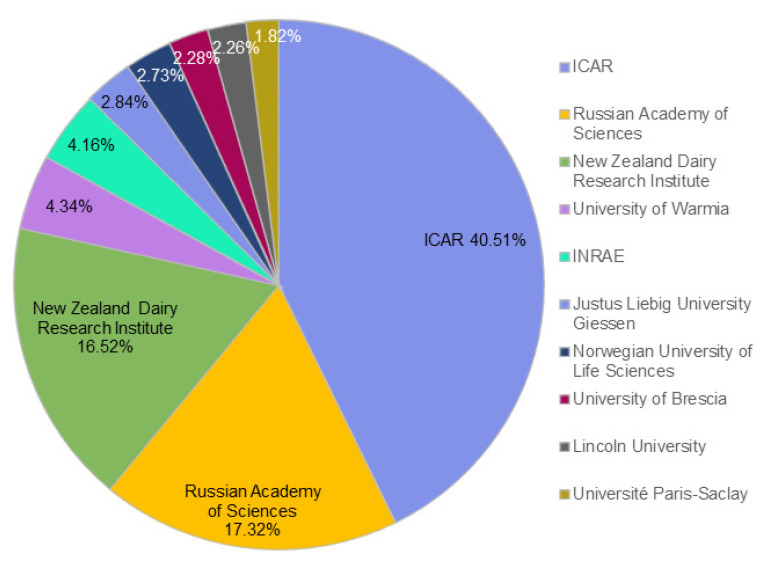
The top 10 most important affiliations in the A2 β-casein research field. The number of publications which referred to A2 β-casein in each affiliation area was relativized with respect to the total number of publications of each affiliation area in order to show the real importance of each affiliation in the A2 β-casein field.

**Table 1 animals-12-01909-t001:** Top 10 most productive affiliations in the A2 β-casein research field.

Position	Affiliations	Records	Country
1	Indian Council of Agricultural Research (ICAR)	37	India
2	l’Institut National de Recherche pour l’Agriculture, l’Alimentation et l’Environnement (INRAE)	23	France
3	Justus Liebig University Giessen	29	Germany
4	University of Melbourne	19	Australia
5	University of Milan	18	Italy
6	University of Warmia	17	Poland
7	Aarhus University	15	Denmark
8	Université Paris-Saclay	15	France
9	Lincoln University New Zealand	13	New Zealand
10	University of Brescia	12	Italy

**Table 2 animals-12-01909-t002:** Distribution of publications by document type.

Document Type	Document Type/Total Documents (N°)	%
Article	502/618	81.2
Review	31/618	5.0
Patent	27/618	4.4
Meeting	21/618	3.4
Other	19/618	3.1
Book	10/618	1.6
Letter	8/618	1.3

**Table 3 animals-12-01909-t003:** Top 10 most productive affiliations in the A2 β-casein research field.

Position	Source Titles	Records	%	5-Years JCR	h-Index	Country
1	Journal of Dairy Science	61	9.87	4.354	191	U.S.A.
2	International Dairy Journal	18	2.91	3.395	140	Netherlands
3	Journal of Dairy Research	16	2.58	2.019	77	U.K.
4	Animal genetics	13	2.10	3.169	81	U.K.
5	Food Chemistry	12	1.94	7.516	262	U.K.
6	Milchwissenschaft Milk Science International	11	1.78	0.247	39	Germany
7	Nutrients	9	1.45	5.719	115	Switzerland
8	Nutrition Journal	7	1.13	3.271	90	U.K.
9	European Journal of Biochemistry ^1^	6	0.97	-	-	U.K.
10	Food Hydrocolloids	6	0.97	9.147	159	Netherlands

JCR = Journal Citation Reports. ^1^ This journal was renamed as FEBS (Federation of European Biochemical Societies) in 2005.

**Table 4 animals-12-01909-t004:** Articles about A2 β-casein and health outcomes with more than 50 citations.

Number of Citations ^1^	Title (Authors)	Year	Affiliation	Type of Study	+/− A2 -Casein ^2^	Reference
177 [177]	*Polymorphism of bovine beta-casein and its potential effect on human health* (Kaminski, S; Cieslinska, A and Kostyra, E)	2007	Univ. Warmia (Poland)	Review	+ (?)	[12]
136 [136]	*Type I (insulin-dependent) diabetes mellitus and cow milk: casein variant consumption* (Elliott, RB; Harris, DP; Hill, JP; Bibby, NJ; Wasmuth, HE)	1999	Univ. Auckland and New Zealand Dairy Research Institute (New Zealand), and Univ. Düsseldorf (Germany)	Field experiment	+	[64]
116 [116]	*Beta-casein A(1), ischaemic heart disease mortality, and other illnesses* (McLachlan, CNS) ^3^	2001	A2 Corp. Ltd. (New Zealand)	Field experiment	+	[65]
83 [83]	*A casein variant in cow’s milk is atherogenic* (Tailford, KA; Berry, CL; Thomas, AC; Campbell, JH)	2003	Univ. Queensland (Australia)	Animal experiment	+	[66]
77 [77]	*Release of beta-casomorphins 5 and 7 during simulated gastro-intestinal digestion of bovine beta-casein variants and milk-based infant formula* (De Noni, I)	2008	Univ. Milan (Italy)	Laboratory experiment	+	[67]
76 [76]	*The A2 milk case: a critical review* (Truswell, AS)	2005	Univ. Sydney (Australia)	Review	−	[68]
73 [73]	*Effects of milk containing only A2 beta casein versus milk containing both A1 and A2 beta casein proteins on gastrointestinal physiology, symptoms of discomfort, and cognitive behavior of people with self-reported intolerance to traditional cows’ milk* (Sun, JQ; Xu, LM; Lu, X; Yelland, GW, Gregory W; Ni, JY; Clarke, AJ) ^3^	2016	Fudan Univ., Shanghai Univ., Medical Centers and Consulting Co. (China), Monash Univ. and RMIT Univ (Australia), and A2 Corp. Ltd. (New Zealand)	Clinical human trial	+	[18]
68 [68]	*Milk Intolerance, Beta-Casein and Lactose* (Pal, S; Woodford, K; Kukuljan, S; Ho, S) ^3^	2015	Curtin Univ., A2 Corp. Ltd. (Australia) and Lincoln Univ. (New Zealand).	Review	+ (?)	[69]
61 [61]	*Health implications of milk containing beta-casein with the A(2) genetic variant* (Bell, SJ; Grochoski, GT; Clarke, AJ) ^3^	2006	Ideasphere Inc. (USA), A2 corp LTD (New Zealand)	Review	+ (?)	[51]
57 [56]	*Comparative effects of A1 versus A2 beta-casein on gastrointestinal measures: a blinded randomised cross-over pilot study* (Ho, S; Woodford, K; Kukuljan, S; Pal, S) ^3^	2014	Curtin Univ., A2 Dairy Products (Australia) and Lincoln Univ. (New Zealand)	Clinical human trial	+	[34]
51 [51]	*Effects of cow’s milk beta-casein variants on symptoms of milk intolerance in Chinese adults: a multicentre, randomised controlled study* (He, M; Sun, JQ; Jiang, ZQ; Yang, YX) ^3^	2017	Beijing, Shanghai and Guangzhou Univ. and Res. Centers, and Chinese Nutrition Society (China) ^2^	Clinical human trial	+	[60]
50 [49]	*Dietary A1 beta-casein affects gastrointestinal transit time, dipeptidyl peptidase-4 activity, and inflammatory status relative to A2 beta-casein in Wistar rats* (Barnett, MPG; McNabb, WC; Roy, NC; Woodford, KB; Clarke, AJ) ^3^	2014	AgResearch, Massey Univ., Lincoln Univ. and A2 Corp. Ltd. (New Zealand).	Animal experiment	+	[70]

^1^ Without self-citing between brackets. ^2^ Positive (+) or Negative (−) result for A2 β-casein outcomes in human (or animal) health; (?) it is indicated that further investigation is needed. ^3^ A2 Corp. Ltd. participated as author or sponsor.

## Data Availability

Data sharing is not applicable.

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
