# Peer review of "Worldwide Research Trends on Milk Containing Only A2 β-Casein: A Bibliometric Study"

_animals, 2022, doi:10.3390/ani12151909_

Round 1
Reviewer 1 Report
The title corresponds to the content of the manuscript
1. It requires supplementing with a specific aim in the abstract
2. The introduction requires rewording, entering a clearly defined goal, and describing the application significance of the research undertaken and the meaning for science
3. Methodology, results described correctly. The summary requires clarification and description of important aspects that result from the study.
Reviewer 2 Report
In this article entitled "Worldwide Research Trends on Milk Containing only A2 β-casein: A Bibliometric Study", the Authors address a current topic of considerable importance to human health, which concerns the role of β-casein in cow milk protein intolerance.
Indeed, among the numerous isoforms of β-casein (at least 12 genetic variants are known), the most frequent are β-casein A1 and A2.
In literature are available many studies that attribute the production of the opioid peptide β-casomorphin-7 (BCM-7), during gastrointestinal digestion, to β-casein A1. In this regard, several human and animal experiments demonstrate the negative effects of β-casein A1 on health.
On contrary, cow's milk in which only β-casein A2 is present does not seem to have these effects; therefore, the dairy industry, also trades milk and milk products labeled "only A2" and "A1 free."
Thus, the Authors conducted and in-depth bibliometric analysis of the research on β-casein A2 using different databases. Many factors were analyzed (such as number of publications per year, research areas, country distribution, affiliations and sources, etc.).
In conclusion, the bibliometric study shows that further efforts are needed to determine the possible implications of this new product in the human population and market.
The review is interesting and easy to read, with many bibliographical citations, and highlights the great work done by the Authors, but needs to be improved in the following minor aspects:
Line 35-37: “Among milk proteins, it can be found three categories depending on their solubility potential: caseins (at around 80%), whey proteins (about 14%) and fat globule membrane proteins [4]”. The sentence is partially true, since in the milk of equidae (horses and donkeys), the Casein Index is lower and ranges on average from 40 to 55%. I suggest rephrasing the sentence.
Line 41-42: “Cow´s milk protein allergy (CMPA) is an adverse immune reaction when the body comes in contact with dietary antigen in cow´s milk [5].” I suggest including a reference about the use of donkey milk as a food to limit the allergy to cow's milk protein allergy.
Line 42-44: Among milk intolerances, the most common one, is lactose intolerance, which is caused by a reduced or absent activity of lactase enzyme leading to lactose malabsorption [6]. “Lactose-free milk” is available on the market, which is a lactose-free commercial dairy product. I suggest to insert a short comment.
Line 50-53: “The most common forms of β-casein among dairy cattle breeds are A1 and A2 [10,11]. The β-casein produced by dairy cows in their milk is controlled by a single gene, located on chromosome 6, called CSN2 52 [12]. A1 β-casein differs from A2 β-casein by having a different amino acid at position ……”. I suggest to introduce the concept of beta casein polymorphism and its known genetic variants; for example: “Dairy cattle have 12 β-casein variants (A1, A2, A3, B, C, D, E, F, H1, H2, I, and G); however, only seven of these (A1, A2, A3, B, C, I, and E) have been detected in European cattle breeds. The A2 variant is considered the oldest variant, from which the others originated via mutation. The most common variants are A1 and A2; the B variant is less common … ”; (source: Sebastiani, C.; Arcangeli, C.; Ciullo, M.; Torricelli, M.; Cinti, G.; Fisichella, S.; Biagetti, M. Frequencies Evaluation of β-Casein Gene Polymorphisms in Dairy Cows Reared in Central Italy. Animals 2020, 10, 252. https://doi.org/10.3390/ani10020252).
Line 193-194: The graphics in Figure 3 must be considerably improved.
Line 294: The term “india” and “sahiwal” ….; rewrite in correct form: The terms “India” and “Sahiwal” .
Line 441-442: In order to make Table 2 more understandable, I suggest that authors also enter the total number of type documents in each row.
for example: Article 530/857 85.7%
line 581-752: I suggest the Authors check the references, in particular the following citations: 2, 21, 27, 35, 36, 44, 70, if they meet the standards indicated by the journal publisher.
Finally, I ask the Authors, but as my personal curiosity, if they have found any references on the dairy suitability of “A2 milk” compared to “A1 milk” (e.g. Milk Clotting Properties).
Reviewer 3 Report
Review “Worldwide Research Trends on Milk Containing only A2 β-casein: A Bibliometric Study”
Purpose/ goal: the authors perform a bibliometric analysis on the literature published on Beta-casein A2 research to determine the state of research worldwide/ worldwide research production (?; manuscript sentence is not clear) in the field of milk or dairy products containing on A2 beta-casein variant.
Firstly, this manuscript needs extensive language editing, many sentences need rephrasing and word use is not correct. This will greatly enhance the readability and quality of the manuscript.
The field of research on A2 vs A1 human health benefits and effects on milk processing is relevant especially because of the aggressive marketing of A2 products.
A bibliometric analysis could give insights in who are the most active players in the field, if there is a possible funding based bias, how much primary research is being published on the topic ( and various relevant sub-topics) as well how many reviews and how much these publications are being cited and thus have the possibility of having an impact on the direction of the research in the field.
By solely reporting on categories and parameters pre-analyzed by WoS leaves may interesting and more insightful questions unanswered. Lost opportunity to give a clear picture of where the research effort stands with regard to A2 beta casein variant.
Although the manuscript reports on number of publication on WoS identified topics more informative would be for instance, analyzing the subfields known to cover the A1/A2 beta casein field of research. This way a better view could be provided on what type of studies are being done and reported on than the very generic WoS derived information. for instance topics pertaining to:
Prevalence of certain genetic variants/protein variants within cow populations and breeds
How genetic/protein variants associate with milk production traits
How variants affect milk processing (physichemical properties)
Human health association (devided in to reviews/perspectives and clinical trails (types of trails can even be subdivided))
This will require manual queries instead of relying on the “categories” defined by WoS which are less informative.
Some of this information can be gleaned from “terms clouds” as presented in the manuscript but overall useful information is lost in generalized overly broad topics and discriptions.
Another aspect that could be further explored:
Which authors are most active in the field
Which publications are most referenced and by what type of publication
I suggest that themMain analysis reported focusses on peer reviewed articles and reviews, as abstract and proceedings etc most likely have overlap with peer reviewed articles and reviews a these often are the initial communication of research findings to be later published as peer reviewed articles or reviews.
A visualization of the overlap of topics would be helpful to understand the extend of the overlap of topic publications and the way these WoS topic overlap in this field
Other aspects that could be explored and might give additional useful insights:
Relationship among publications
Co-citation analysis
Relationship among cited publications
Foundational themes (this seems to be addressed)
Co-authorship analysis
Bibliographic coupling
Methods:
A figure with a flow chart of search approach and number of results should be included.
Only using WoS because it is easier to subtract the information and analyze ( analysis are reported upon search by WoS) it is not a real good scientific justification, how much important information is not included by ignoring other information sources?
Calrify if all data bases represented in WoS are included in the search or just WoS science “core collection” or something else?
Search for beta-casein might also include assays using casein for bacterial enzyme activity so this is way broader than what is probably included in the A2/A1subset.
A supplementary file with a table listing the included A1/A2 publications would be appropriate
Not sure why Beta-casein search is included in figure 2 and analysis
Related to Figure 1
It would be helpful to use inset to show CSN variant data as the scale for general beta-casein and A1/A2 publications is so large.
Also it would be helpful to address increase of A1/A2 related publications in numbers and as percentage of all beta casein publications this will give a better idea how much this field has gained in importance. ( maybe broken down by type of publication.
Then it would be helpful to know what the proportion of publication on different subtopics/fields is within the A1/A2 publications (see previous comment)
Review vs research data publications on the topic of A1/A2 and human health (what type of human health issues are being studied)
Furthermore, identifying who the Authors are that are mainly publishing within certain topics/sub-topics. This is probably captured with the analysis of the countries and institutions of origin, but it gives a very different view/interpretation of the research effort. If it is only one lab/ person publishing or if there are many labs involved in research topic in a country.
The sponsor information is very relevant to identify possible bias in the reporting of result and in the weight that is given to impact and importance of publications.
Another consideration is of it is just a few authors publishing different reviews or perspective etc articles on the A1/A2 and human health outcomes or primary-results based publications.
Scientific impact (# of citations of a publication) should also address self-citation even though that is incorporated in some of the citation index calculations
Patents is a good topic to also include as that sets the stage of where innovation is and can go
(not sure if need such an extensive explanation of the source data)
Think the conclusion is that the majority of the publications on A1/A2 beta casein currently are related to the identification of the genetic variants in animal populations (mostly dairy cattle) and the association of such variants to milk production traits and physiochemical properties of milk containing these variants. The research of impact of A1/A2 on human systemic review show limited evidence of adverse effects of A1 vs A2.
Reviewer 4 Report
Dear authors,
Some comments about this manuscript, in addition to those in the attached file are the following:
- I would like that you can explain better why only 71 references worldwide about this research topic were selected? Did you tried another database, such as, for example Scopus?
- Did you found any information about the possible consequences of removal of A1 beta-casein from milk?

Round 2
Reviewer 3 Report
I appreciate the replies from the authors regarding content of the manuscript and am satisfied with the changes made regarding content of the manuscript.
However, there are still too many English language issues that need to be addressed to permit publication of the manuscript as is.
It is absolutely essential that a native English speaker or preferably a professional scientific English editor edits the current version before publication is considered.
I have indicated the sections/ sentences that I encountered(and noticed) in the first 5 pages of the manuscript that need to be addressed, to exemplify the need for English language editing throughout the manuscript:
Introduction:
Line 34-36: numerous issues
Line 40: ..intolerances are increasingly [2,5]. …are Increasing…?
Line 47-49: numerous issues (see below)
Those condictiones (not quite sure what it refers to),
…researchers have been studied… > researchers have studied
… the possible implication… > the possibility …?
… that the composition of the protein fraction of beta casein may have in the manifestation…
>>incomplete sentence
Line 52: …and, within them,… & … represents a 30% >> and beta casein accounts for 30% of the casein fraction?
Line 60 and 63: … associated to… >> associated with
Line 70: .. may be interesting among milk protein intolerance population. >> May be of interest to the population with milk protein intolerance. (?)
Line 80: …since they are investing a lot of resources on it…. >> …investing… in it …(?) or devoting… to it….(?)
Methods
Line 87-89: First of all… obtained through WoS…. >> numerous issues
Line 98: On this point… >> at this point
Line 99-100: …. Would have conducted to low number of publications
Line 108: restricted from…>> restricted to…
Line128-129: Finally, the most 50 cited articles focusing on…..>> Finally, the 50 most cited articles….
Results and discussion
Line137-138: …a general increase in the number of publications was under gone, but with several fluctuations throughout years. >> ….a general increase in the number of publications occurred ( took place) with several fluctuations throughout the years.
Line 138: Stands up 1990s with … >> the 1990’s standout with a sudden….
Line139-140: After that decade, the trend is of a progressive increase, with… >>not proper English
Figure 1 legend/ line145: ..since the year was not ended… >> had not ended
Line150: … a considerably much lower number….. >>a considerably lower number….
Line152:… not until the final of the 1990s.... >> not until the end of the 1990s
Line 152: number of written documents .. not sure you need written here
Line 154: … when it was experienced an increase,… >> not proper English
Line 155: …were developed… >> were published ??
Line 157:.. on this topic were reached up.. >> ..on this topic was reached..
Line 161:… publications on the beta -casein research field..>> publications in the beta-casein research field
Line 173: .. documents in beta-casein research field..>> ..in the beta casein research field..
( in general it is: the beta casein research field, to be adjusted all through)
Line 183-184: The rest research areas showed in the figure 2, include less than 200 records on each…>> many issues, my best guess: The remaining research areas shown in figure 2 included less than 200 record each.
Line 185-186:
Line190: resided in…
Dimension in the context of a country is incorrect word use: “size” would be correct
Line 343: dimensions >>> size…
line 411: low dimension >>> small size
a scientific journal is not a magazine
Many, many more but I am not an editor.
Line 400-402 (and 403-404) appears to be close quotes from reference 65 and IMO not correctly interpreted from what is discussed in that article, nor is this explanation needed here.
Not sure referring to “importance” of an “affiliations” based on the number of publications from a particular institutions (affiliation) is the right way of phrasing it.
Author Response
Ref: animals-1762921
Dear Dr. Devon Liu,
We have revised the manuscript with the reference number animals-1762921.
Thank you very much for the suggestions proposed to improve the clarity of the text. The English language have been checked by professional English editor.
Regarding the Reviewer 3 comments, in reference to Lines 400-402 and 403-404, we agree with the reviewer and this paragraph have been removed to improve the clarity of the text (Lines 414-418). In addition, the reviewer stated that “Not sure referring to “importance” of an “affiliations” based on the number of publications from a particular institutions (affiliation) is the right way of phrasing it.”. In Table 1, the productivity of the affiliations was analyzed considering the number of publications, however, to study the importance of an affiliation, the number of publications referred to A2 β-casein in each affiliation area was relativized with respect the total number of publications of each affiliation area in order to appreciate the real importance of each affiliation on A2 β-casein field.
We have uploaded the revised version of the document. The changes have been marked up using the “Track Changes” function.
Sincerely,
Dr. Olaia Urrutia